# Combustion Mechanism of Alloying Elements Cr in Ti-Cr-V Alloys

**DOI:** 10.3390/ma12193206

**Published:** 2019-09-30

**Authors:** Lei Shao, Yayu Wang, Guoliang Xie, Hongying Li, Jiashuai Xiong, Jiabin Yu, Guangyu He, Jinfeng Huang

**Affiliations:** 1State Key Laboratory for Advanced Metals and Materials, University of Science and Technology Beijing, No. 30, Xueyuan Road, Beijing 100083, China; shaoleiustb@163.com (L.S.); wyy79b@163.com (Y.W.); leon_xq@126.com (G.X.); deliahy@163.com (H.L.); g20179010@xs.ustb.edu.cn (J.X.); 13161879524@163.com (J.Y.); 2Science and Technology on Plasma Dynamics Laboratory, Air Force Engineering University, No.1, Changle Road, Xi’an 710043, China; hegy_22@126.com

**Keywords:** combustion mechanism, alloying elements, Ti-Cr-V alloys

## Abstract

The combustion velocity and the mechanism for a series of Ti-Cr-V alloys with different chemical compositions are studied by a promoted ignition combustion test corresponding to different oxygen pressures to investigate the influence of alloying elements, such as Cr and V, on combustion behavior. The microstructures and composition distributions of the alloying elements in the reaction and oxide areas are observed and analyzed. The thermogravimetry analysis results show that the oxidation mass gain decreases with the increasing Cr content, and the oxidation resistance obviously increases from 10 Cr to 20 Cr. The combustion velocity decreases with increasing Cr content, and it is concluded that elevated Cr content can effectively retard the flame propagation velocity. Importantly, for the Ti-Cr-V alloys, the Cr and V elements accumulate in the melting zone and reduce the heat created by combustion.

## 1. Introduction

Titanium alloys are widely used in advanced aero engines because of their excellent high specific strength, corrosion resistance and heat resistance [1,2]. However, titanium alloys burn easily due to their low thermal coefficient and high combustion heat, a phenomenon called titanium fire [3,4]. Since the titanium alloys have been used in aero engines, the titanium fire accidents have occurred frequently. Once it occurs, the engines and titanium parts are burned out in 2–20 s, with almost no means of extinguishing the fire.

To fundamentally solve this problem, many researchers have tried to develop new burn-resistant titanium alloys. Currently, the design of burn-resistant titanium alloys is based on the three following principles. The first is interrupted oxygen transmission [5,6,7,8,9,10]. The addition of chromium and vanadium elements in pure titanium forms a dense oxide film to prevent Ti from oxidizing. The V element transports excessive heat away from the melting zone due to the volatilization of V_2_O_5_, which prevents heat accumulation at the combustion front. Therefore, it efficiently avoids combustion. The next involves thermodynamics [11,12,13]. Adding V, Cr, Mo and other elements that have a low combustion heat to Ti can reduce its adiabatic combustion temperature. The third involves a reduction of the fiction [14,15,16,17,18]. Adding Cu, Al and other elements to pure Ti forms the Ti_2_Cu phase that has a low melting point. The dry friction between titanium parts is changed into wet friction with liquid lubrication from the Ti_2_Cu phase, and the heat of friction decreases sharply. On the other hand, the addition of Cu to titanium alloys increases their thermal conductivity and disperses the heat rapidly to avoid local temperature rise, which makes it difficult to reach the melting point and inhibits their combustion.

Many scholars have studied titanium fires from many different perspectives, however, the effect of alloying elements on the combustion behavior of titanium alloys have been rarely reported. Thus, to effectively prevent titanium fires, the fireproof performance of a series of Ti-Cr-V alloys was evaluated by the promoted ignition combustion (PIC) test [19] in this study. The burning characteristics, such as the burning velocity and burned sample structure, were studied to discuss the effect of element composition during the burning process. 

## 2. Experimental

### 2.1. Material Preparation

Alloy ingots with a nominal composition of Ti-25V-10Cr, Ti-25V-15Cr, and Ti-25V-20Cr were fabricated by vacuum induction melting using a mixture of pure metals of titanium, vanadium and chromium (purity > 99.99 wt%), then normalized at 850 °C for 1 h followed by water quenching to room temperature. The ingot was hot forged and hot rolled into a 5 mm thick plate at 850 °C. The ingots were cut into rods with a diameter of 3.2 mm and the length of 40 mm by line cutting and surface polishing.

### 2.2. Experimental Process 

The PIC tests were carried out using a piece of equipment designed according to the American Society for Testing Materials (ASTM) G-124 standard [19] as shown in Figure 1. A copper wire with a 1 mm diameter was attached to the end of the sample as the resistance wire to provide sufficient energy for ignition. The chamber was pumped to a vacuum of 10^−1^ to 10^−2^ Pa, and then, the gaseous oxygen was pumped into the setting pressures. The resistance wire was electrically heated and ignited. The entire combustion process was recorded by a video camera, as shown in Figure 2. Once the sample was ignited, it did not stop burning until all of the sample was burned away. To reserve a portion of the samples to do the subsequent analysis, the argon was pumped into force extinguishing the combustion process. The combustion velocity of the specimen was determined, and the microstructures and chemical compositions of the reaction areas were analyzed. The PIC tests were carried out at the oxygen pressures from 0.1 to 1.0 MPa to investigate the influence of the microstructures on the combustion behaviors.

### 2.3. Characterization Methods 

The specimens were cut into two pieces along the longitudinal section, ground, polished and etched in an HF:HNO_3_:H_2_O = 1:3:5 solution for microstructure observations. The phase formation in the longitudinal sections of the samples was determined by X-ray diffraction (XRD) on a Huber-2 goniometer (Cu-Kα radiation, TTR3, Rigaku, Japan). This was followed by microstructure characterization using an optical microscope and scanning electron microscopy (SEM) with a Zeiss Supra55 operated at 20 keV, along with energy-dispersive spectrometry (EDS). The non-isothermal oxidation experiments were conducted by thermogravimetric analyzer (TGA) (SDT Q600, NSK, Japan) with an accuracy of 0.01 mg. The specimens with a weight of 20 mg were cut and ultrasonically cleaned and put in the alumina crucible. The specimens were heated from room temperature to 1300 °C at the heating rate of 10 °C/min under a flowing gas mixture of nitrogen (80 mL/min) and oxygen (20 mL/min).

## 3. Results and Discussion

### 3.1. Combustion Characteristics

Generally, the combustion reaction of a metal is a violent oxidation reaction between the metal and oxygen. Figure 2 is an in suit picture during the combustion. The titanium alloy burns violently with a dazzling white light when the titanium alloy sample heated by a resistance wire reaches the ignition temperature. The titanium alloy is melted by the combustion heat and forms the L1 phase [20,21]. Because the solid solubility of the melting alloy is much higher than that of the solid alloy, a large amount of oxygen atom solutes is present in the melting alloy and reacts with it. The alloy then transforms into the L2 phase. As the reaction continues, the oxygen atom in the atmosphere diffuses through the L2 phase and reacts with the L1 phase at the interface of L1/L2 phases continuously, releasing a high combustion heat to move the solid/liquid interface forward. The volume of the melted droplet grows and finally drops due to gravity; it is predominantly composed of melted oxide and the alloy before oxidation. The remaining sample continues to burn until it is consumed. 

The burning velocity of the samples at different oxygen pressures is shown in Figure 3a. The PIC tests at the same oxygen pressure were repeated three times to ensure the reliability of the experimental data. As can be seen from the picture, the combustion velocity of these three alloys is sensitive to oxygen pressure. When the oxygen pressure is high, the combustion velocity is fast. Therefore, the combustion velocity of the alloys increases at high oxygen pressures that have a high oxygen content. The combustion velocity of the three alloys is different at the same oxygen pressure because of the different Cr contents, and the combustion velocity decreases with the increasing Cr content. At lower pressures, the change in absolute values of velocity is about the same for all samples, but the velocity with 20% Cr grows slower than with 15 and 10%. This variation of velocity agrees with those by Zhao [22,23,24,25], who concluded that the combustion of Ti alloys is affected by the adiabatic flame temperatures (AFT). The AFT can be used to characterize the thermal effect, and the combustion mechanism can be studied by calculating the AFT. From the perspective of adiabatic flame temperatures, the AFT of Ti is much higher than that of Cr and V, and the Cr and V elements have no significant impact on AFT when the Ti content is above 60 wt%. However, the AFT decreases when the Ti content is below 60 wt%, and this result suggests that the AFT decreases when the content of vanadium and chromium is above 40 wt%.

Generally, the non-isothermal oxidation characteristic can reflect the combustion property of an alloy to an extent and is directly indicated by the oxidation weight gain. The non-isothermal oxidation carves of alloys from room temperature to 1300 °C are plotted in Figure 3b. All samples exhibit a similar tendency with temperature, and the masses of the three types of alloys have no obvious change before 700 °C. The Ti-V-10Cr alloy mass gains rapidly when the temperature reaches 800 °C, suggesting that the samples begin to vigorously oxidize, but the Ti-V-15Cr and Ti-V-20Cr alloys oxidize vigorously until 900 °C. From the non-isothermal oxidation carves, it is found that the mass gain of Ti-V-20Cr is less than that of Ti-V-10Cr and Ti-V-15Cr at the same oxidation temperature. Overall, the above results show that the Ti-V-20Cr alloy has better combustion resistance than that of the Ti-V-10Cr and Ti-V-15Cr alloys.

### 3.2. Microstructure and Composition

The samples consist of three different zones after combustion, namely the heat-affected zone, melting zone and oxide zone, from the top to the bottom of the sample, as shown in Figure 4a. In the oxide zone, three different types of morphology can be distinguished in Figure 4b, the gray base, an area with white spheroids and an area with fine white particles. The quantitative composition obtained by EDS is listed in Table 1. The gray base consists of Ti and O and appears to be titanium oxide. The composition of the areas with white spheroids and fine white particles is the same, mainly consisting of Cr, V and O. XRD was conducted to investigate the phase changes after combustion. As illustrated in Figure 5a, the combustion products of Ti-V-10Cr, Ti-V-15Cr and Ti-V-20Cr are the same. Combined with EDS, the gray base may be a mixture of TiO, TiO_2_ and Ti_2_O_3_ and the white phases are a mixture of Cr_2_O_3_ and V_2_O_5_. According to Figure 4b, the oxide zone is dense and different from that of the TC4 and TC11, which contains many holes and cracks in the oxide zone. In the melting zone, two different microstructures are found. In the bottom melting zone, close to the oxide zone, there is a chrysanthemum-like structure, and in the top melting zone, close to the heat-affected zone, there is a cellular-dendrite structure (Figure 4c). To analyze the variation of the elements proportion in the reacting zone, EDS line scanning was performed along the melting zone and oxide zone in the post-combustion Ti-Cr-V alloys (Figure 5b). In the melting zone, the content of V and Cr remains constant, whereas their content decreases dramatically at the interface. In the heat-affected zone, as shown in Figure 4d, it can be seen that the grain boundaries coarsen and melt. This result shows that the solid/liquid interface moves forward along the grain boundaries in Ti-Cr-V alloys.

### 3.3. The Role of Alloying Elements

According to the above results, the difference in the combustion properties of these three alloys is caused by different Cr contents. The TGA results show that the alloy begins to oxidize drastically when the resistance wire heats the alloy to a critical temperature. Then, the alloy releases a tremendous amount of heat, and the titanium alloy begins to combust when the temperature reaches the ignition point. According to the literature [26], the combustion flame temperature of the titanium alloy is approximately 2700 °C, which is much higher than its melting point, so the titanium alloy will be melted. As is well known, the dissolved oxygen in the melting alloy is higher than that in the solid alloy, and the combustion reaction is very violent until all the sample is burned away. In the study of high-temperature oxidation of metal materials, the standard formation of free energy of metallic oxide (ΔG^θ^) is used to judge the possibility of metal oxidation. The value of ΔG^θ^ can be obtained directly from the oxygen-potential diagram. It is known that the more negative the value of the ΔG^θ^ is, the more stable the oxide and the stronger the affinity between the metal and oxygen. From the oxygen-potential diagram, it is clear that the ΔG^θ^ of Ti is the most negative followed by V and Cr. Consequently, Ti reacts with O preferentially and leads to the formation of a porous Ti oxide layer. As the reaction continues, Cr and V diffuse outward and react with O and form Cr_2_O_3_ and V_2_O_5_ spheroids among the Ti oxide layer, as shown in Figure 4b. The composite oxide layer is compact and integrated, so it is difficult for oxygen to diffuse inward to react with titanium. During the combustion reaction, titanium reacts with oxygen preferentially because of the strong affinity between Ti and O. Thus, the Cr and V elements accumulate in the melting zone, as shown in Figure 5b and Figure 6. The richness of Cr and V is described quantitatively in Figure 7 and Table 2. Compared with the three alloys, Ti-25V-20Cr alloy shows the highest element enrichment of Cr and V. When the flame extends to the zone enriched with Cr and V, a small amount of heat is released since the combustion heat of Cr and V (2608 cal/g ad 3637cal/g) is lower than that of Ti (4717 cal/g) [27], and when the flame extends to the zone which is enriched with Cr and V, less heat will be released. It is well known that the burning velocity is the velocity that the solid/liquid interface moves forward, and the lower the combustion heat is, the slower the velocity of the solid/liquid surface moves forward and the slower the burning velocity. Among the three alloys (Ti-25V-10Cr, Ti-25V-15Cr and Ti-25V-20Cr), the higher the Cr content is, the higher the enrichment and the lower the combustion heat, so the decreasing order of the burning velocity is Ti-25V-10Cr, Ti-25V-15Cr and then Ti-25V-20Cr.

## 4. Conclusions

This work studied the combustion mechanisms for Ti-Cr-V alloys by PIC test at different oxygen pressures. The following conclusions can be drawn:The post-combustion Ti-Cr-V alloys consist of three different zones, namely the heat-affected zone, melting zone and oxide zone.The oxide zone is composed of titanium oxides, Cr_2_O_3_ and V_2_O_5_, moreover, the Cr_2_O_3_/V_2_O_5_ composites are spherical and they randomly disperse in the oxide zone.The combustion velocity decreases with increasing Cr content, and the variation in the combustion velocity is related to the enrichment of Cr and V in the melting zone. This enrichment phenomenon in the melting zone reduces the heat created by combustion and further slows the combustion velocity.

## Figures and Tables

**Figure 1 materials-12-03206-f001:**
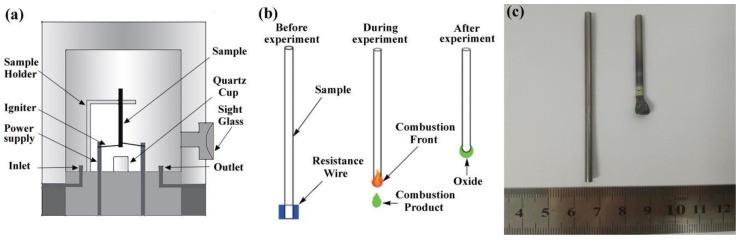
(**a**) Promoted ignition-combustion testing equipment, (**b**) Schematic diagram of ignition combustion samples, (**c**) samples after combustion.

**Figure 2 materials-12-03206-f002:**
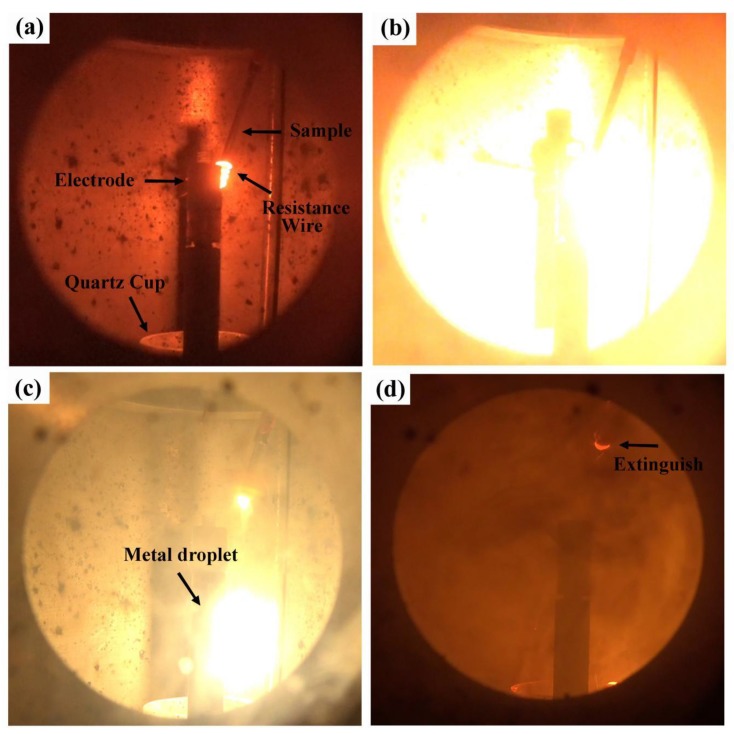
In suit picture during the combustion of Ti-V-Cr alloy samples. (**a**) Heating by resistance wire, (**b**) ignition, (**c**) combustion and (**d**) extinguished sample.

**Figure 3 materials-12-03206-f003:**
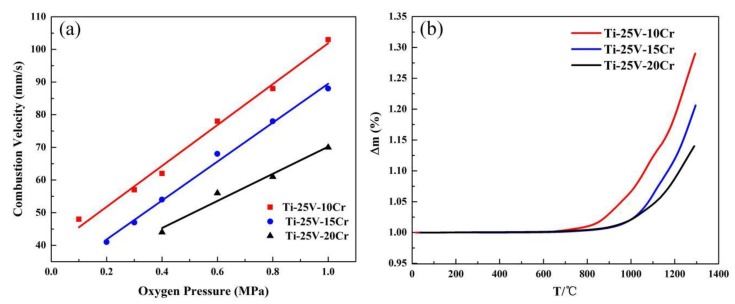
(**a**) The combustion velocity of specimens with the increasing oxygen pressures from 0.1 MPa to 1.0 MPa. (**b**) Mass gain curves of Ti-Cr-V alloys duo to non-isothermal oxidation from room temperature to 1300 °C.

**Figure 4 materials-12-03206-f004:**
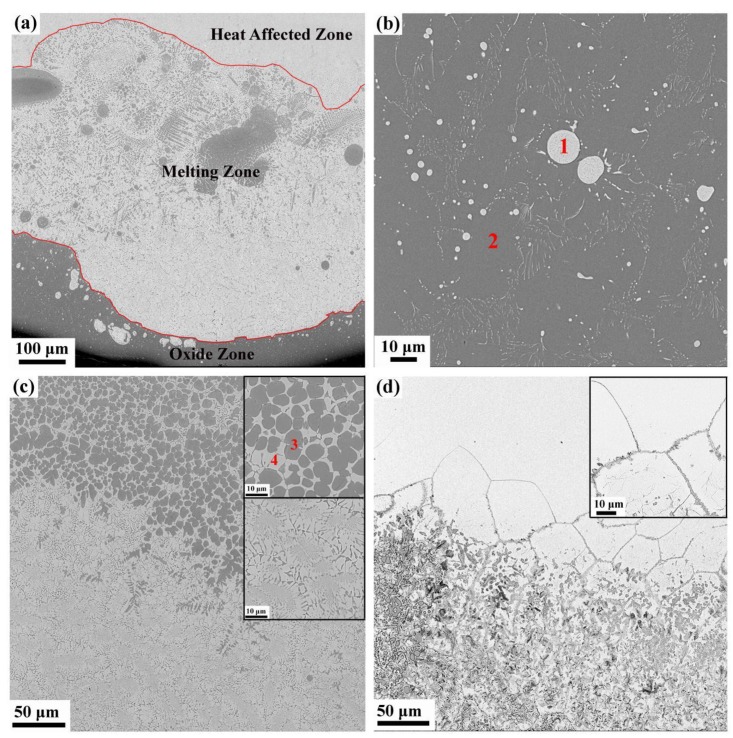
The SEM photographs, (**a**) the morphology of entire reaction area, consisting of heat-affected zone, melting zone and oxide zone, (**b**) oxide zone, and the 1 and 2 marks are two different phases in oxide zone and their compositions are shown in Table 1, (**c**) melting zone, and the 3 and 4 marks are two different phases in melting zone and their compositions are shown in Table 1, (**d**) heat-affected zone.

**Figure 5 materials-12-03206-f005:**
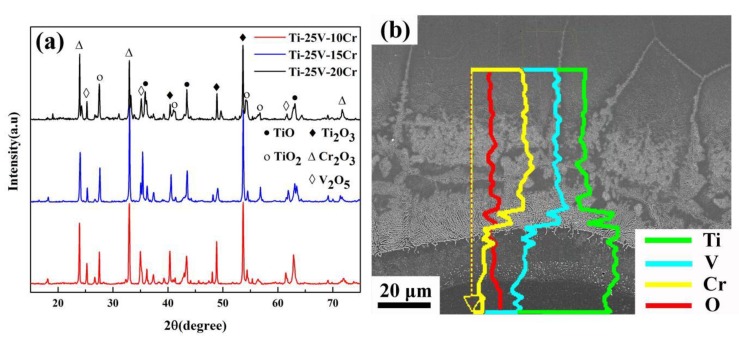
(**a**) XRD analysis results of Ti-Cr-V alloys after combustion, (**b**) line scanning curves of EDS analysis across the melting zone and oxide zone.

**Figure 6 materials-12-03206-f006:**
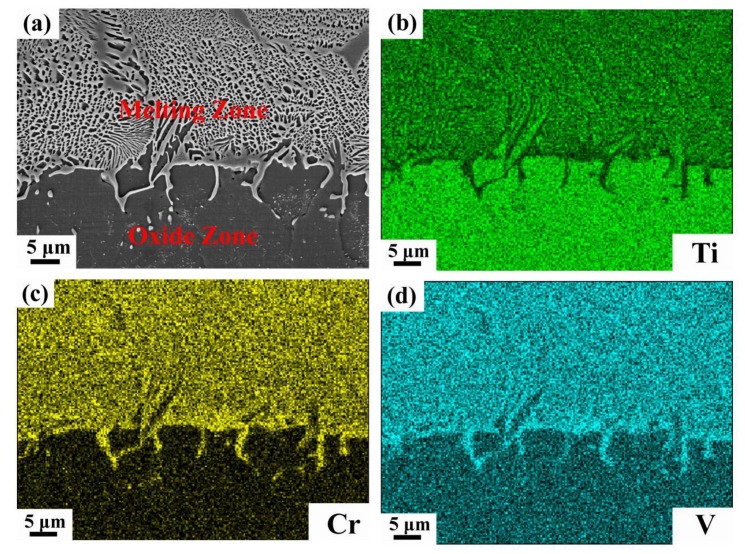
SEM photograph of typical microstructure of oxide zone and melting zone (**a**), and the corresponding mapping-scan of EDS analysis, containing (**b**–**d**) for Ti, Cr and V atomic distributions, respectively.

**Figure 7 materials-12-03206-f007:**
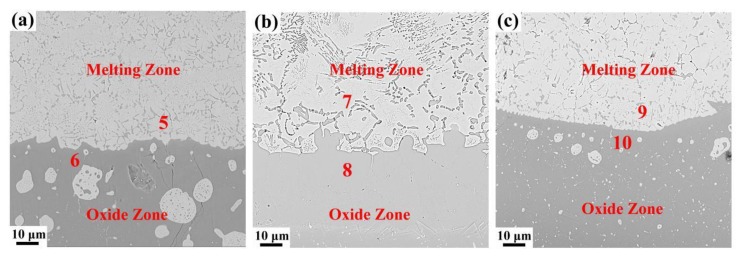
The interface between melting zone and oxide zone, (**a**) Ti-25V-10Cr, (**b**) Ti-25V-15Cr, (**c**) Ti-25V-20Cr.

**Table 1 materials-12-03206-t001:** Chemical compositions of different areas in oxide and melting zone.

Region	Composition			
	Ti	Cr	V	O
1 (at%)	7.47	30.01	39.06	23.45
1 (wt%)	8.35	36.43	46.45	8.76
2 (at%)	61.42	0.10	2.21	36.27
2 (wt%)	80.82	0.14	3.09	15.95
3 (at%)	65.94	4.20	9.06	20.80
3 (wt%)	75.72	5.23	11.07	8.76
4 (at%)	7.47	30.01	39.06	15.84
4 (wt%)	11.22	37.49	45.71	5.58

**Table 2 materials-12-03206-t002:** Chemical compositions of different areas in oxide and melting zone in each test alloy.

Region	Composition			
	Ti	Cr	V	O
5 (at%)	27.71	16.61	23.13	32.55
5 (wt%)	34.11	22.21	30.29	13.39
6 (at%)	54.11	--	1.02	44.87
6 (wt%)	77.09	--	1.55	21.37
7 (at%)	23.85	22.86	25.29	28
7 (wt%)	28.08	31.68	31.68	11.02
8 (at%)	63.33	0.53	3.02	33.12
8 (wt%)	80.99	0.74	4.11	14.16
9 (at%)	7.65	25.20	32.78	34.37
9 (wt%)	9.40	33.63	42.86	14.11
10 (at%)	52.24	0.11	1.14	46.52
10 (wt%)	75.58	0.17	1.76	22.50

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
