# Peer review of "Combustion Mechanism of Alloying Elements Cr in Ti-Cr-V Alloys"

_materials, 2019, doi:10.3390/ma12193206_

Round 1
Reviewer 1 Report
The paper shows interesting results on the effect of Cr content on the mechanism and normal burning velocity in oxygen of Ti-V-Cr compositions. However, before publishing the paper it is worth to describe the global brut reaction and indicate the richness or the equivalence ratio corresponding to each test.
Finally, the English of the paper needs to be improved by a native English-speaking editor.
Author Response
Dear Editor and Reviewers
Thank you very much for giving us this opportunity to revise our manuscript. We appreciate the editor and reviewers very much for their constructive comments and suggestions on our manuscript entitled “Combustion mechanism of alloying elements Cr in Ti-Cr-V alloys” submitted to “materials”.
We have studied reviewers’ comments carefully. According to the reviewers’ detailed suggestions, we have made a careful revision on the original manuscript. All revised portions are marked in red in the revised manuscript which we would like to submit for your kind consideration. Our response of the comments is enclosed at the end of this letter. We also have our manuscript modified by a qualified company to help us improve the English writing.
Thanks again for your kindness and advices.
If you have any question about this paper, please don’t hesitate to contact us.
Sincerely yours,
Jinfeng Huang
Response to Reviewer 1:
Thanks for your comments on our paper. We have revised the paper according to your comments. The detaled revisions are listed as follows:
The paper shows interesting results on the effect of Cr content on the mechanism and normal burning velocity in oxygen of Ti-V-Cr compositions. However, before publishing the paper it is worth to describe the global brutreaction and indicate the richness or the equivalence ratio corresponding to each test. Finally, the English of the paper needs to be improved by a native English-speaking editor.
Response: Thanks for your kind suggestions. The global burning reaction is described in the Line 88 to 99. The richness of each alloy is described in the Line 173 to 175 and shown in Fig.7 and Table 2. The English is polished by the professional company.
Reviewer 2 Report
The paper reqiures significant language improvement. The number of corrections that need to be made is very large. The far from exhaustive list of corrections is proposed below. The text requires major revision.
Language corrections:
line 16: ’it is conclude’
concluded
lines 18-19: ’reduce the created heat by combustion’.
reduce the heat created by combustion
lines 29-30: ’Come so far, the design of burn resistance titanium alloys is
based on the three principles below.’
’currently, ... is based’ , ’principles described below’
line 36: ’ fictional heating’
frictional
lines 47 - 49: ’The burning characteristics, e.g., burning duration, burning
velocity, burned sample structure is performed to discuss the effect of element
composition (chromium and vanadium) during burning process and derive the
burn resistant property of the alloys.
should be ’are performed’; also, not characteristics are performed, but mea-
surements. The sentence should be rephrased.
line 67: ’ to do the subsequence analysis,’
subsequent
line 67-68: ’ argon was pumped in to forced extinguish the combustion
process.’
to force extinguishing?
line 81: ”Generally, The combustion reaction””
lower case ’the’
line 82: ”in-suit’
in situ
line 82-84: ’The titanium alloy burns violently with a dazzling white light
when the temperature of titanium alloy sample reaches to the ignition temper-
ature heated by resistance wire’.
”...when titanium alloy sample heated by a resistance wire reaches the igni-
tion temperature...”
lines 85-87:’A large amount of oxygen atom solutes in the melting alloy and
reacts with it, and turn into L2 phase, because the solid solubility of melting
alloy is much higher than solid alloy’.
Should be corrected and rephrased.
line 87-89: ”With the reaction continues, the oxygen atom in atmosphere
through L2 phase and reacted with L1 phase at the interface of L1/L2 contin-
uously, releasing huge combustion heat to push the solid/liquid interface move
forward”
”oxygen atoms diffuse”? ”reacts with L1 phase”
The sentence should be rephrased.
line 90: ’The melted droplet is getting bigger and bigger’,
Why not just ’grows’, or to emphasize estimate the growth estimate the
growth quantitatively. ’bigger and bigger’ is impressive and poetic, but non-
informative.
line 93: ’The burning velocity of the samples at different oxygen pressures
is listed in Fig. 3(a)’
’shown’ in place of ’listed’
line 97: ’This can understand easily that alloys combust faster and more
sufficient in high oxygen pressure with higher content of oxygen.’
Should be rephrased. ’can be understood’ What is ’combust more sufficient’ ?
line 145: ”Function of alloying elements of Cr and V”
The role?
line 148: ’the alloy begin’
’the alloy begins’
line 148: ’ the resistance wire heat alloy’
’wire heats’
line 150: ’ when the temperature reach to ignition point’
’temperature reaches the ignition point’
line 152: ’As we all known, the dissolved oxygen in melting alloy is higher
than solid alloy’,
’As is well known’
lines 156-157: ’And the value of ∆Gθ can be get directly from the oxygen-
hetapotential diagram. It is know that the more negative of the ∆Gt ’
’can be obtained’, ’it is known’, ’the more negative value of’
line 163: ’has little cavities and cracks. so it is is different for oxygen to
diffuse inward through the dense oxide layer to react with titanium.’
’cracks, so it is ...’ Should be rephrased. What is different?
line 164-167: ”During the combustion reaction, a lot of titanium atom from
the melting zone diffuse to combustion interface to react with oxygen, cause the
strong affinity between Ti and O.”
’a lot of titanium atoms’ , ’because of the strong affinity’ ?
lines 169 - 171: ’As we know, the burning velocity is the velocity of solid/liquid
interface move forward, and the less the combustion heat the slower the velocity
of solid-liquid surface move forward as will as the burning velocity’
Should be rephrased
line 182: ”The combustion velocity decrease with the increasing”
’velocity decreases’
”Fig.2. In-suit picture during the combustion of Ti-V-Cr alloy samples”.
in situ
---------------------
The questions regarding the content:
line 14: 'The results of DSC ...'
DSC or TGA?
Line 47-49: "The burning characteristics, e.g., burning duration, ... is performed "
No burning duration results were reported in this paper.
Table 1
What are zones 1, 2, 3, 4? While zones 1 and 2 are shown in Fig. 4b, no references are made to zones 3 and 4. What are these zones?
lines 81-90: description of combustion mechanism - how information about mechanism of combustion was obtained? Is it taken from the literature? Then the reference is needed.
lines 100-104: 'And the combustion velocity shows little difference when the Cr content increase from 10 wt% to 15 wt%, but when Cr content reach to 20 wt % , the combustion velocity decrease obviously. This observation suggests that only the Cr content exceeds a certain value, it can play a role in combustion.'
It seems that Fig. 3 contradicts this statement. The figure shows that at lower pressures, the change in absolute values of velocity is about the same for all samples, but the velocity with 20% Cr grows slower than with 15 and 10%. Anyway, figure 3 is hardly supporting the strong statement that 'only the Cr content exceeds a certain value, it can play a role in combustion'.
lines 107-109: "be studied by calculating the AFM. From the view of adiabatic flame temperatures, the AFM of Ti is much higher than Cr and V, and the Cr and V element has no significant impact on AFM when the Ti content is above 60 wt%."
What is AFM?
lines 154-160: 'In the study of high temperature oxidation of metal materials, the
standard formation of free energy of metallic oxide (ΔGθ) is used to judge the possibility of
metal oxidation. And the value ofΔG can be get directly from the oxygen-potential diagram. It is know that the more negative of the ΔGθ, the more stable of the oxide and the stronger affinity between the metal and oxygen. From the oxygen-potential diagram, it is clear that the ΔGθ of Ti is the most negative followed by V and Cr. Consequently, Ti
reacts with O preferentially and leads to form the porous Ti oxide layer. '
What exactly the free energy is used for? To see if the reduction of oxide occurs? Isn't oxidation of titanium at high (as well as low temperatures) the trivial fact?
lines 160: 'leads to form the porous Ti oxide layer'
'leads to formation of the porous oxide'..
This statement requres clarification. Above, in line 134 the authors stated that 'According to Fig. 4(b), the oxide zone is dense and continuous'. How is it inferred that the oxide is porous? Is it the authors hypothesis or well known fact (reference)?
lines 178-179: 'The Ti-Cr-V alloys consist three different zones after combustion, from top to bottom , the zones are heat affected zone, melting zone and oxide zone respectively'.
Must be rephrased. Not 'after combustion'. It should be changed to something like 'zones surrounding the combustion front (or area?)'.
Fig. 1
Small font. Also, probably, in all the figures the font could be increased.
Fig 2
'extinguish, 'metal droplet' marks need clarification, 'extinguish' arrow points at the bright spot, which does not look like 'extinguish'. Would be more clear to extend the caption: a). ignition b). burning ... (d) burned (or extinguished) sample and so on.
Fig 4(b):
Would help if 1 and 2 marks were clarified in the caption, and, possibly, referred to table 1. These zones are described in the text, but it would make it easier for understanding to briefly define these marks in the caption.
Fig 6.
Also, would make it easier if it was indicated which zone is melt, which zone iz oxide.
Author Response
Dear Editor and Reviewers
Thank you very much for giving us this opportunity to revise our manuscript. We appreciate the editor and reviewers very much for their constructive comments and suggestions on our manuscript entitled “Combustion mechanism of alloying elements Cr in Ti-Cr-V alloys” submitted to “materials”.
We have studied reviewers’ comments carefully. According to the reviewers’ detailed suggestions, we have made a careful revision on the original manuscript. All revised portions are marked in red in the revised manuscript which we would like to submit for your kind consideration. Our response of the comments is enclosed at the end of this letter. We also have our manuscript modified by a qualified company to help us improve the English writing.
Thanks again for your kindness and advices.
If you have any question about this paper, please don’t hesitate to contact us.
Sincerely yours,
Jinfeng Huang
Response to Reviewer 2:
Thanks for your comments on our paper. We have revised the paper according to your comments. The grammar and spelling errors have also been corrected. The detaled revisions are listed as follows:
Language corrections:
line 16: ’it is conclude’ concluded.
Response: Thanks for your kind suggestions. This mistake has been corrected in the revised manuscript.
lines 18-19: ’reduce the created heat by combustion’.reduce the heat created by combustion.
Response: Thanks for your kind suggestions. This mistake has been corrected in the revised manuscript.
lines 29-30: ’Come so far, the design of burn resistance titanium alloys isbased on the three principles below.’ ’currently, ... is based’ , ’principles described below’.
Response: Thanks for your kind suggestions. This sentence has been rephrased according to your advice.
line 36: ’ fictional heating’frictional.
Response: Thanks for your kind suggestions. This mistake has been corrected in the revised manuscript.
lines 47 - 49: ’The burning characteristics, e.g., burning duration, burningvelocity, burned sample structure is performed to discuss the effect of element composition (chromium and vanadium) during burning process and derive the burn resistant property of the alloys. should be ’are performed’; also, not characteristics are performed, but measurements. The sentence should be rephrased.
Response: Thanks for your kind suggestions. This mistake has been corrected in the revised manuscript and the sentence has been rephrased.
line 67: ’ to do the subsequence analysis,’subsequent.
Response: Thanks for your kind suggestions. This mistake has been corrected in the revised manuscript.
line 67-68: ’ argon was pumped in to forced extinguish the combustion’ to force extinguishing?
Response: Thanks for your kind suggestions. This mistake has been corrected in the revised manuscript.
line 81: ”Generally, The combustion reaction”lower case ’the’.
Response: Thanks for your kind suggestions. This mistake has been corrected in the revised manuscript.
line 82: ”in-suit’in situ.
Response: Thanks for your kind suggestions. This mistake has been corrected in the revised manuscript.
line 82-84: ’The titanium alloy burns violently with a dazzling white lightwhen the temperature of titanium alloy sample reaches to the ignition temperature heated by resistance wire’. ”...when titanium alloy sample heated by a resistance wire reaches the ignition temperature...”
Response: Thanks for your kind suggestions. This sentence has been rephrased according to your advice.
lines 85-87:’A large amount of oxygen atom solutes in the melting alloy andreacts with it, and turn into L2 phase, because the solid solubility of melting alloy is much higher than solid alloy’. Should be corrected and rephrased.
Response: Thanks for your kind suggestions. This mistake has been corrected and the sentence has been rephrased in the revised manuscript.
line 87-89: ”With the reaction continues, the oxygen atom in atmospherethrough L2 phase and reacted with L1 phase at the interface of L1/L2 continuously, releasing huge combustion heat to push the solid/liquid interface move forward”. ”oxygen atoms diffuse”? ”reacts with L1 phase” The sentence should be rephrased.
Response: Thanks for your kind suggestions. This mistake has been corrected and the sentence has been rephrased in the revised manuscript.
line 90: ’The melted droplet is getting bigger and bigger’,Why not just ’grows’, or to emphasize estimate the growth estimate the growth quantitatively. ’bigger and bigger’ is impressive and poetic, but noninformative.
Response: Thanks for your kind suggestions. The phrase “bigger and bigger ” is replaced by “grows” in the revised manuscript.
line 93: ’The burning velocity of the samples at different oxygen pressuresis listed in Fig. 3(a)’ ’shown’ in place of ’listed’
Response: Thanks for your kind suggestions. The word “listed” is replaced by “shown” in the revised manuscript.
line 97: ’This can understand easily that alloys combust faster and moresufficient in high oxygen pressure with higher content of oxygen.’ Should be rephrased. ’can be understood’ What is ’combust more sufficient’ ?
Response: Thanks for your kind suggestions. This sentence has been rephrased according to your advice.
line 145: ”Function of alloying elements of Cr and V”The role?
Response: Thanks for your kind suggestions. The word “Function” is replaced by “The role”.
line 148: ’the alloy begin’’the alloy begins’.
Response: Thanks for your kind suggestions. This mistake has been corrected in the revised manuscript.
line 148: ’ the resistance wire heat alloy’’wire heats’.
Response: Thanks for your kind suggestions. This mistake has been corrected in the revised manuscript.
line 150: ’ when the temperature reach to ignition point’’temperature reaches the ignition point’.
Response: Thanks for your kind suggestions. This mistake has been corrected in the revised manuscript.
line 152: ’As we all known, the dissolved oxygen in melting alloy is higherthan solid alloy’, ’As is well known’.
Response: Thanks for your kind suggestions. The phrase “As we all know” is replaced by “As is well known” in the revised manuscript.
lines 156-157: ’And the value of ∆Gθ can be get directly from the oxygen-hetapotential diagram. It is know that the more negative of the ∆Gt ’’can be obtained’, ’it is known’, ’the more negative value of’.
Response: Thanks for your kind suggestions. This mistake has been corrected in the revised manuscript.
line 163: ’has little cavities and cracks. so it is is different for oxygen todiffuse inward through the dense oxide layer to react with titanium.’’ cracks, so it is ...’ Should be rephrased. What is different?
Response: Thanks for your kind suggestions. This sentence has been rephrased in the revised manuscript. It should be “difficult” rather than the “different”. We feel sorry for this mistake.
line 164-167: ”During the combustion reaction, a lot of titanium atom fromthe melting zone diffuse to combustion interface to react with oxygen, cause the strong affinity between Ti and O.”’a lot of titanium atoms’ , ’because of the strong affinity’ ?
Response: Thanks for your kind suggestions. This sentence has been rephrased in the revised manuscript.
lines 169 - 171: ’As we know, the burning velocity is the velocity of solid/liquidinterface move forward, and the less the combustion heat the slower the velocity of solid-liquid surface move forward as will as the burning velocity’ Should be rephrased.
Response: Thanks for your kind suggestions. This sentence has been rephrased in the revised manuscript.
line 182: ”The combustion velocity decrease with the increasing”’velocity decreases’.
Response: Thanks for your kind suggestions. This mistake has been corrected in the revised manuscript.
”Fig.2. In-suit picture during the combustion of Ti-V-Cr alloy samples”.in situ.
Response: Thanks for your kind suggestions. This mistake has been corrected in the revised manuscript.
The questions regarding the content:
line 14: 'The results of DSC ...'DSC or TGA?
Response: Thanks for your kind suggestions. It should be TGA and this mistake has been corrected in the revised manuscript.
Line 47-49: "The burning characteristics, e.g., burning duration, ... is performed "No burning duration results were reported in this paper.
Response: Thanks for your kind suggestions. The burning duration is deleted in the revised manuscript.
Table 1What are zones 1, 2, 3, 4? While zones 1 and 2 are shown in Fig. 4b, no references are made to zones 3 and 4. What are these zones?
Response: Thanks for your kind suggestions. Regions 1, 2, 3, 4 represent the different phases in specimen. Regions 1 and 2 represent the two different phases in oxide zone and regions 3 and 4 represent another two phases in melting zone. Regions 3 and 4 are shown in the first illustration in Fig. 4c.
lines 81-90: description of combustion mechanism - how information about mechanism of combustion was obtained? Is it taken from the literature? Then the reference is needed.
Response: Thanks for your kind suggestions. We obtained the information about mechanism of combustion from the video and literature. The relevant literatures are added in the revised manuscript.
lines 100-104: 'And the combustion velocity shows little difference when the Cr content increase from 10 wt% to 15 wt%, but when Cr content reach to 20 wt % , the combustion velocity decrease obviously. This observation suggests that only the Cr content exceeds a certain value, it can play a role in combustion.'
It seems that Fig. 3 contradicts this statement. The figure shows that at lower pressures, the change in absolute values of velocity is about the same for all samples, but the velocity with 20% Cr grows slower than with 15 and 10%. Anyway, figure 3 is hardly supporting the strong statement that 'only the Cr content exceeds a certain value, it can play a role in combustion'.
Response: Thanks for your kind suggestions. The original statement is not accurate and it is corrected in the revised manuscript.
lines 107-109: "be studied by calculating the AFM. From the view of adiabatic flame temperatures, the AFM of Ti is much higher than Cr and V, and the Cr and V element has no significant impact on AFM when the Ti content is above 60 wt%."
What is AFM?
Response: Thanks for your kind suggestions. It should be AFT (adiabatic flame temperature) not the ATM. We feel sorry for this mistake.
lines 154-160: 'In the study of high temperature oxidation of metal materials, thestandard formation of free energy of metallic oxide (ΔGθ) is used to judge the possibility of metal oxidation. And the value of ΔGqcan be get directly from the oxygen-potential diagram. It is know that the more negative of the ΔGθ, the more stable of the oxide and the stronger affinity between the metal and oxygen. From the oxygen-potential diagram, it is clear that the ΔGθ of Ti is the most negative followed by V and Cr. Consequently, Ti reacts with O preferentially and leads to form the porous Ti oxide layer. '
What exactly the free energy is used for? To see if the reduction of oxide occurs? Isn't oxidation of titanium at high (as well as low temperatures) the trivial fact?
Response: Thanks for your kind suggestions. The standard formation of free energy of oxide (ΔGθ) represents the stability of oxide. And a more stable metallic oxide means the metal is more likely to react with oxygen. The ΔGθ of titanium oxide is lower than that of vanadium and chromium at any temperature.
lines 160: 'leads to form the porous Ti oxide layer''leads to formation of the porous oxide'..
This statement requres clarification. Above, in line 134 the authors stated that 'According to Fig. 4(b), the oxide zone is dense and continuous'. How is it inferred that the oxide is porous? Is it the authors hypothesis or well known fact (reference)?
Response: Thanks for your kind suggestions. The titanium oxide layer is porous and the compounded oxide in oxide zone (include vanadium oxide and chromium oxide) is dense and continuous.
lines 178-179: 'The Ti-Cr-V alloys consist three different zones after combustion, from top to bottom , the zones are heat affected zone, melting zone and oxide zone respectively'.Must be rephrased. Not 'after combustion'. It should be changed to something like 'zones surrounding the combustion front (or area?)'.
Response: Thanks for your kind suggestions. The sentence is rephrased in the revised manuscript .
1Small font. Also, probably, in all the figures the font could be increased.
Response: Thanks for your kind suggestions. The fonts are increased in all the figures in the revised manuscript.
Fig 2 'extinguish, 'metal droplet' marks need clarification, 'extinguish' arrow points at the bright spot, which does not look like 'extinguish'. Would be more clear to extend the caption: a). ignition b). burning ... (d) burned (or extinguished) sample and so on.
Response: Thanks for your kind suggestions. The Fig. 2(d) is replaced by a more appropriate figure and the caption is extended in the revised manuscript.
Fig 4(b):Would help if 1 and 2 marks were clarified in the caption, and, possibly, referred to table 1. These zones are described in the text, but it would make it easier for understanding to briefly define these marks in the caption.
Response: Thanks for your kind suggestions. The 1, 2, 3, 4 marks are clarified in the caption and referred to table 1.
Fig 6.Also, would make it easier if it was indicated which zone is melt, which zone is
Response: Thanks for your kind suggestions. And the marks are added in Fig. 6 in the revised manuscript.
Reviewer 3 Report
1. It is necessary to specify the equipment (for DSC, XRD, SEM, EDS) with which the analysis was performed.
2. For DSC analysis, it is necessary to indicate the heating rate and composition of the gas atmosphere in which the study was carry out.
Author Response
Dear Editor and Reviewers
Thank you very much for giving us this opportunity to revise our manuscript. We appreciate the editor and reviewers very much for their constructive comments and suggestions on our manuscript entitled “Combustion mechanism of alloying elements Cr in Ti-Cr-V alloys” submitted to “materials”.
We have studied reviewers’ comments carefully. According to the reviewers’ detailed suggestions, we have made a careful revision on the original manuscript. All revised portions are marked in red in the revised manuscript which we would like to submit for your kind consideration. Our response of the comments is enclosed at the end of this letter. We also have our manuscript modified by a qualified company to help us improve the English writing.
Thanks again for your kindness and advices.
If you have any question about this paper, please don’t hesitate to contact us.
Sincerely yours,
Jinfeng Huang
Response to Reviewer 3:
Thanks for your comments on our paper. We have revised the paper according to your comments. The detaled revisions are listed as follows:
It is necessary to specify the equipment (for DSC, XRD, SEM, EDS) with which the analysis was performed.
Response: Thanks for your kind suggestions. The analytical equipment is specified in the revised manuscript.
For DSC analysis, it is necessary to indicate the heating rate and composition of the gas atmosphere in which the study was carry out.
Response: Thanks for your kind suggestions. The heating rate and composition of the gas atmosphere are indicated in the revised manuscript.
Round 2
Reviewer 2 Report
Dear authors! Interesting work, the paper is improved a lot. Just two typos remain to be corrected: line 86 and Fig. 2 caption: change, please, 'in suit' to 'in situ'.